# Schnider and Eleveld Models for Propofol Target-Controlled Infusion Anesthesia: A Clinical Comparison

**DOI:** 10.3390/life13102065

**Published:** 2023-10-16

**Authors:** Federico Linassi, Paolo Zanatta, Leonardo Spano, Paolo Burelli, Antonio Farnia, Michele Carron

**Affiliations:** 1Department of Pharmaceutical and Pharmacological Sciences, Università Degli Studi di Padova, Via Marzolo 5, 35131 Padova, Italy; 2Department of Anesthesiology and Critical Care, Treviso Regional Hospital, AULSS 2 Marca Trevigiana, Piazzale Ospedale 1, 31100 Treviso, Italy; paolo.zanatta1@aulss2.veneto.it (P.Z.); antoniofarnia@gmail.com (A.F.); 3Department of Medicine—DIMED, Section of Anesthesiology and Intensive Care, University of Padova, 35100 Padova, Italy; leonardo.spano@studenti.unipd.it (L.S.); michele.carron@unipd.it (M.C.); 4Department of Breast Oncologic Surgery, Treviso Regional Hospital, AULSS 2 Marca Trevigiana, Piazzale Ospedale 1, 31100 Treviso, Italy; paolo.burelli@aulss2.veneto.it

**Keywords:** intravenous anesthesia, propofol, pharmacokinetics, intraoperative monitoring, awareness, intraoperative complications

## Abstract

Background: Various pharmacokinetic/pharmacodynamic (PK/PD) models have been developed to accurately dose propofol administration during total intravenous anesthesia with target-controlled infusion (TIVA-TCI). We aim to clinically compare the performance of the Schnider model and the new and general-purpose Eleveld PK/PD model during TIVA-TCI. Methods: We conducted a prospective observational study at a single center, enrolling 78 female patients, including 37 adults (aged < 65 years) and 41 elderly patients (aged ≥ 65 years). These patients underwent breast surgery with propofol-remifentanil TIVA-TCI guided by the bispectral index (BIS) for depth of anesthesia monitoring (target value 40–60) and the surgical plethysmographic index (SPI) for antinociception monitoring (target value 20–50) without neuromuscular blockade. The concentration at the effect site of propofol (CeP) at loss of responsiveness (LoR) during anesthesia maintenance (MA) and at return of responsiveness (RoR), the duration of surgery and anesthesia (min), the time to RoR (min), the propofol total dose (mg), the deepening of anesthesia events (DAEs), burst suppression events (BSEs), light anesthesia events (LAEs) and unwanted spontaneous responsiveness events (USREs) were considered to compare the two PK/PD models. Results: Patients undergoing BIS-SPI-guided TIVA-TCI with the Eleveld PK/PD model showed a lower CeP at LoR (1.7 (1.36–2.25) vs. 3.60 (3.00–4.18) μg/mL, *p* < 0.001), higher CePMA (2.80 (2.55–3.40) vs. 2.30 (1.80–2.50) μg/mL, *p* < 0.001) and at RoR (1.48 (1.08–1.80) vs. 0.64 (0.55–0.81) μg/mL, *p* < 0.001) than with the Schnider PK/PD model. Anesthetic hysteresis was observed only in the Schnider PK/PD model group (*p* < 0.001). DAEs (69.2% vs. 30.8%, *p* = 0.001) and BSEs (28.2% vs. 5.1%, *p* = 0.013) were more frequent with the Eleveld PK/PD model than with the Schnider PK/PD model in the general patient population. DAEs (63.2% vs. 27.3%, *p* = 0.030) and BSEs (31.6% vs. 4.5%, *p* = 0.036) were more frequent with the Eleveld PK/PD model than with the Schnider PK/PD model in the elderly. Conclusions: The Schnider and Eleveld PK/PD models impact CePs differently. A greater incidence of DAEs and BSEs in the elderly suggests more attention is necessary in this group of patients undergoing BIS-SPI-guided TIVA-TCI with the Eleveld PK/PD than with the Schnider model.

## 1. Introduction

Target-controlled infusion (TCI) is a widely used computer-controlled method for achieving and maintaining a stable concentration at the effect site of propofol (CeP) based on physiological parameters, such as the patient’s age, weight, height, and gender during total intravenous anesthesia (TIVA) [1,2]. The TIVA-TCI system uses a population-derived pharmacokinetic/pharmacodynamic (PK/PD) model incorporated in a computer-controlled infusion pump to determine the initial bolus dose and, subsequently, to reach and maintain the target CeP [1,2].

Various PK/PD models have been developed to accurately dose propofol administration during TIVA-TCI [1,2]. There is no evidence to support the use of one PK/PD model in preference over another and all PK/PD models have proven reliable in clinical practice [3]. One limitation of widely employed PK/PD models for propofol is that they were developed mainly using healthy volunteers or patients and not with challenging populations, such as children and obese or elderly patients whose body composition or elimination mechanisms may be structurally different from those of the lean adult patient population [2,3].

The Eleveld propofol PK/PD model (from here on referred to as “model” for convenience) was developed for a broad population range as TIVA-TCI for children, adults, elderly subjects, and obese adults undergoing general anesthesia [4]. However, in a prospective study comparing the predictive performance of the Eleveld model with other models in a TCI system, for PK, the Eleveld model showed a clinically acceptable performance in children, adults, and obese adults undergoing general anesthesia, but not in older subjects, where a great bias (27%) was observed [5].

The aim of this study was to investigate, primarily, the performance of the Schnider and Eleveld models comparing the CeP at loss of responsiveness (LoR) during anesthesia maintenance and at the return of responsiveness (RoR), and, secondarily, the incidence of unwanted events during anesthesia in adult patients undergoing TIVA-TCI with one of the two models.

## 2. Materials and Methods

### 2.1. Study Design

This prospective observational study, involving adult patients undergoing oncologic breast surgery at the Treviso Regional Hospital AULSS 2 Marca Trevigiana Piazzale Ospedale 1, 31100, Treviso (Italy), where TIVA-TCI is routinely used as the preferred anesthesia technique for patients undergoing surgery under general anesthesia, was approved by the local Institutional Ethical Committee (approval number 681/CE Marca). All subjects participating in the trial provided written informed consent. The trial was registered prior to patient enrollment at ISRSCTN registry (ISRCTN41934206; principal investigator: Linassi Federico; link to trial registry: https://www.isrctn.com/ISRCTN41934206, date of registration: 25 October 2022).

### 2.2. Study Population

From 30 October to 23 December 2022, female (≥18 years) patients undergoing surgical procedures (quadrantectomy or mastectomy) for breast cancer were consecutively recruited. Oncologic breast surgery is a well-standardized procedure typically performed on female patients with fewer health issues and a lower risk of bleeding, hypotension, and hemodynamic instability compared to other surgeries in our hospital.

The STROBE (Strengthening the Reporting of Observational Studies in Epidemiology) checklist was followed.

Patients with an American Society of Anesthesiologists (ASA) physical status classification of >3; a body mass index (BMI, kg/m^2^) of ≥35 (due to the limitations of Schnider TCI model in managing obese patients); neurodegenerative (e.g., Alzheimer’s disease, Parkinson’s disease) or cerebrovascular (e.g., transient ischemic attack, stroke) diseases; psychiatric disorders; and respiratory (i.e., asthma, chronic obstructive pulmonary disease), cardiovascular (e.g., chronic heart failure, coronary artery disease, arrhythmia, peripheral vascular disease), kidney (e.g., end-stage renal disease), and liver (e.g., cirrhosis) diseases were excluded due to their potential to affect cognitive functions and the individual’s response to anesthesia. Patients in continuous therapy with anxiolytics or antidepressant drugs or with a history of alcohol or psychoactive drug abuse were not enrolled due to brain effects and risks associated with differences in enzyme induction. Patients requiring perioperative anxiolysis or intraoperative vasoactive drugs or receiving neuromuscular blocking agents during surgery were also excluded to avoid the risk of underestimating unwanted spontaneous responsiveness events (USREs). Patients who were deaf or hard of hearing at preoperative evaluation were not enrolled.

The patient assignment to either the Schnider model group or the Eleveld model group was by chance, independently of patient scheduling, based on alternation between Schneider and Eleveld model users within the framework of a daily 6 h shift rotation of the anesthesiologists involved in the study, who adopted a priori the model they were used to or most familiar with during the study period [1,3].

### 2.3. General Anesthesia

Before anesthesia induction, continuous electrocardiography, pulse oximetry, and noninvasive blood pressure measurements were conducted and an intravenous line in the arm for TIVA-TCI was established. A bilateral electrode strip (COVIDien IIC, Mansfield, MA, USA) was placed on the patient’s forehead and linked to an XP monitor (Monitor BIS Module A-2000 Revision 3.12) to evaluate the level of consciousness and depth of anesthesia using the bispectral index (BIS). The BIS was calculated using a computer algorithm that analyzes the frontal lobe electroencephalogram (EEG) and displays it as a number on the BIS view monitor. BIS values range from 0 to 100. A value of 0 represents an absence of brain activity, and 100 represents an awake state. BIS values between 40 and 60 represent adequate general anesthesia for surgery, values less than 40 represent a deep hypnotic state, and values greater than 60 represent a light hypnotic state [6].

Concerning analgesia, the nociception–antinociception balance was monitored using the surgical plethysmographic index (SPI) [7,8]. The SPI is a normalized score that is based on a photoplethysmographic analysis of the pulse wave and the heartbeat interval. The values of the SPI range from 0 to 100. Maintaining a value between 20 and 50 is generally recommended during general anesthesia in adult patients [7,8,9] and this was the target in this study.

Induction and maintenance of anesthesia were performed using TIVA-TCI. CeP and Ce of remifentanil (CeR) were achieved using a uSP6000 syringe pump infusion system (Arcomed AG, Kloten, Switzerland), using the Schnider [10,11] or Eleveld [4] models for propofol and the Minto [12,13] model for remifentanil. The starting CeP was set at 1 μg/mL and continued at CeP increments of 0.5 μg/mL up to CeP 3–4 μg/mL (for patients <50 years of age) or 2–3 μg/mL (for patients >50 years of age) [1,4]. An investigator blinded to group allocation checked for LoR, defined as spontaneous eye closure associated with the inability to execute simple verbal commands (loss of connected spontaneous responsiveness)—for example, ‘Anne! Open your eyes!’—every 10 s after the propofol infusion was started. The TIVA-TCI was then adjusted by changes of 0.5 μg/mL at intervals of ≥1 min until the BIS reached the target value of 40–60 [14] after anesthesia induction and whenever a change of BIS target was observed during anesthesia maintenance. CePMA1 and CePMA2 were the CePs registered at the beginning and end of the BIS-SPI-guided anesthesia maintenance, respectively. Specifically, CePMA1 was the CeP observed after having evaluated BIS and SPI five times at one-minute intervals simultaneously to confirm the steady state, while CePMA2 was the last value of CeP during anesthesia maintenance before stopping TIVA-TCI at the end of the surgery.

To blunt noxious stimulation [15] without affecting LoR [16], the starting CeR was set at 0.8 ng/mL before starting propofol infusion and, after LoR, it was adjusted for intraoperative analgesia to a target SPI of 20–50 through changes of 0.5 ng/mL at intervals of ≥1 min until reaching the suggested SPI range [1,17,18,19].

After LoR, a laryngeal mask was placed, and volume-controlled protective lung ventilation was started.

At the end of the surgery, 30 mg ketorolac tromethamine, 1 g paracetamol, and 4 mg ondansetron were given intravenously for pain and postoperative nausea and vomiting (PONV) prophylaxis, respectively. Then, TIVA-TCI was targeted to a CeP of 0 μg/mL and a CeR of 0 ng/mL. With the return of spontaneous ventilation at RoR, defined as spontaneous eye-opening, obtained without providing auditory or tactile stimulation during the awakening process, associated with the ability to execute simple verbal commands after eye-opening (return of connected spontaneous responsiveness)—for example, ‘Anne! Shake my hand!’—the laryngeal mask was removed. The patient was then transferred to the post-anesthesia care unit.

### 2.4. Clinical Endpoints and Variables

Data were collected regarding age (years); BMI (kg/m^2^); American Society of Anesthesiologists (ASA) physical status; CeP (μg/mL) at LoR during anesthesia maintenance (CePMA1 and CePMA2) and at RoR; duration of surgery and anesthesia (min); time to RoR, defined as time to RoR after turning CeP = 0 at the end of the surgery; and propofol total dose (mg).

The primary endpoint was to evaluate the difference in CePs during anesthesia performed with the Schnider and Eleveld models. The secondary endpoints were to evaluate the difference in the incidence of unwanted events during anesthesia performed with the two PK/PD models: lightening of anesthesia event (LAE), defined as BIS > 60 [6]; deepening of anesthesia event (DAE), defined as BIS < 40 [6]; unwanted spontaneous responsiveness event, defined as any involuntary movement (i.e., movement of extremities) or somatic reaction (i.e., coughing, chewing, grimacing, breathing against a ventilator, or inadequate ventilation because of vocal cord closure) in combination with a significant hemodynamic response (i.e., tachycardia (>100 bpm) and hypertension (mean arterial pressure > 120% of baseline or mean arterial pressure ≥ 100 mmHg)) during anesthesia maintenance [7,8,9]; a burst suppression event (BSE), which is defined as a burst suppression ratio > 0. To avoid false positives occurring at low BSR values, according to the literature, BSR > 5% was chosen. Thus, a BSE was therefore defined as a BSR >5% episode, which means at least 3 s of suppressed activity in 1 min of EEG activity [9,20,21].

An investigator who was not involved in delivering anesthesia to patients recorded the data of the variables on a paper data-collection form. This investigator was in charge of electronically recording and storing the BIS data and any unwanted events during anesthesia. Data about CePs were collected from the display of the TIVA-TCI at the respective time points.

### 2.5. Statistical Analysis

Estimating the mean and standard deviation [22] from the median and range of the Median Absolute Performance Error between the Eleveld model and the Schnider model in adults [5], the sample size was based on the following assumptions: a significant Mean Absolute Performance Error difference of 4% between two models in adults, a standard deviation of 6.3%, type I error equal to 0.05, and type II error equal to 0.2 (power [1 − β] = 0.8). Considering these assumptions, the sample size was calculated as 78 patients, equally divided between the Eleveld model group (39 patients) and the Schnider model group (39 patients).

Descriptive analysis was used to summarize the characteristics of the sample. Median and interquartile range (IQR) values were reported for continuous variables. A two-tailed Mann–Whitney U test was used to compare continuous variables between the Eleveld model group and the Schnider model group. Categorical data were reported as an absolute number and as a percentage (%) and compared using an χ^2^ or Fisher’s exact test when more than 20% of cells have expected frequencies < 5 [23]. Friedman’s two-way ANOVA and the Wilcoxon signed-rank test were used to determine whether there was a statistically significant difference in CePs across the four time points and between paired data, respectively, in each model group. Bravais–Pearson’s correlation test was used to determine the strength and direction of association between CePs and age, height, body weight, and BMI. Correlation coefficients (CCs), 95% confidence intervals (95%CI), and *p*-values were determined. To determine the relationships between the dependent categorical (e.g., dichotomous) variable (i.e., unwanted events during anesthesia) and one or more independent categorical variables (i.e., demographic characteristics, PK/PD model), a multiple logistic regression analysis was performed to calculate odds ratios (ORs) with 95%CI [24]. Multicollinearity was assessed using variance inflation factors. The Akaike information criterion and backward/forward stepwise regression analysis were used to choose the best model. Statistical significance was set at *p*-value < 0.05. All statistical analyses were performed using R version 3.4.0 (21 April 2017).

## 3. Results

A total of 86 women undergoing breast oncologic surgery were enrolled. Eight patients had to be excluded from the study. Thus, 78 patients, comprising 37 adults and 41 elders, were considered (Figure 1).

Demographic characteristics and data about all patients—adults and elders—are reported in Appendix A. CePs differed significantly across the four time points within group analysis (*p* < 0.001), mainly due to a significant difference between CeP at RoR and other CePs (*p* < 0.001) in both model groups. Unwanted events during anesthesia were observed in 53.8% of the total patients. An LAE was observed in 3.8%, a USRE in 3.8%, a DAE in 50%, and a BSE in 16.7% of the total cases (Appendix A). Beyond age and ASA, there were no significant differences between the adults and the elders (Appendix A).

### 3.1. Comparison between the Total Population of Patients in the Schnider and Eleveld Model Groups

There were no differences in demographic characteristics, propofol dose, duration of surgery, duration of anesthesia, and time to RoR between the two model groups of patients (Table 1).

At LoR, CeP, and BIS were significantly lower in the Eleveld model group than in the Schnider model group (Table 1, Figure 2). During maintenance of anesthesia, CePs were significantly higher in the Eleveld model group than in the Schnider model group (Table 1, Figure 2). At RoR, CeP was significantly higher in the Eleveld model group than in the Schnider model group (Table 1, Figure 2). The ΔCeP, defined as the difference between CeP LoR and CeP ROR, was significantly different between the models (Table 1), with the hysteresis effect significant only for the Schnider model. No other significant differences were observed (Table 1).

The incidence of total unwanted events during anesthesia was higher in the Eleveld model group than in the Schnider model group (74.4% vs. 33.3% of patients; *p* = 0.001). A significant difference in the incidence of unwanted events during anesthesia related to a deepening—not a lightening—of anesthesia level was observed between the two groups (Table 1).

Concerning the impact of demographic characteristics on CePs, only a significant correlation was observed between age and CeP at RoR in the Schnider model group (CC [95%CI] = −0.493 [−0.7–(−0.21)], *p* = 0.001) (Appendix A).

At logistic regression analysis, the Eleveld model, when compared with the Schnider model, was associated with an increased risk of DAEs (OR [95%CI]: 5.06 [1.94–13.20], *p* < 0.001) and BSEs (OR [95%CI]: 7.27 [1.49–35.40], *p* = 0.014), and a reduced risk of LAEs and USREs (OR [95%CI]: 0.04 [0.01–0.12], *p* < 0.001) during TIVA-TCI (Appendix A).

### 3.2. Comparison between the Schnider Model Group and the Eleveld Model Group in Adults and in Elders

There were no significant differences in demographic characteristics, total propofol dose, and the duration of surgery and anesthesia between the two model groups of patients in both adults and elders (Table 1).

At all the time points, CePs differed significantly; they were lower at LoR and higher during maintenance of anesthesia and at RoR in the Eleveld model group than in the Schnider model group in both adults and elders (Table 1). A significant hysteresis effect was observed in the Schnider model group in both adults and elders (Table 1). At LoR and RoR, BIS differed significantly in adults but not in elders; it was lower at LoR and higher at RoR in the Eleveld than in the Schnider model group (Table 1). Moreover, the time to LoR did not differ significantly between the two model groups in both adults and elders, while the time to RoR was significantly higher in the Eleveld model group than in the Schnider model group in adults but not in elders (Table 1).

The Incidence of total unwanted events during anesthesia was higher in the Eleveld than in the Schnider model group in adults (85% vs. 35% of patients; *p* = 0.003) and in elders (63.2% vs. 31.8% of patients; *p* = 0.063).

No significant difference in the incidence of unwanted events related to a lightening of anesthesia level was observed between the two model groups in both adults and elders (Table 1). A significant difference in the incidence of unwanted events related to a deepening of anesthesia level was observed between the two model groups for DAEs in both adults and elders and for BSEs only in elders (Table 1).

## 4. Discussion

Differences in the PK/PD profile between the Schnider and Eleveld models may justify our findings. The Schnider model, a three-compartment mammillary model, has fixed values for the volume of distribution in the central compartment (V1) and in the second peripheral volume of distribution (V3); adjusts the volume of distribution in the peripheral compartment (V2) for age; and uses height, weight, and sex as covariates of metabolic clearance [3]. The Eleveld model has no fixed values for V1, V2, or V3, and demographic variables such as age, weight, height, and sex were identified as covariates to improve the prediction performance of the model [2]. In general, CeP reaches the targeted propofol concentration more quickly in a patient when a PK/PD model with a smaller V1 or decreased clearance is used than it does when the model has a larger V1 or clearance [25]. The plasma effect site equilibration rate constant (ke0) further affects CeP [25]. A PK/PD model with a high ke0 produces a fast equilibration between plasma and effect site concentration, which results in a higher CeP at LoR and lower CeP at RoR compared to that in a PK/PD model that has a low ke0 [26,27,28]. The fact that ke0 is higher and fixed in the Schnider model (0.456 min^−1^) [2] and lower (0.146 min^−1^) but scaled by (weight/70)-0.25 in the Eleveld model [4] may support the difference in CePs between the two model groups observed in our study.

The phenomenon in which anesthetic induction occurs at higher drug concentrations than the concentrations at emergence is called hysteresis [29,30]. It is attributed to neural inertia, the central nervous system’s inherent resistance to change between consciousness and unconsciousness based on discontinuous behaviors of sensory and motor-related thalamocortical networks at anesthesia induction and emergence [29]. However, hysteresis is not solely attributable to neural inertia [29], but also to the kinetics of drug equilibration [30]. Using a PK/PD model with a fast ke0 resulted in different CePs at LoR and at RoR, thus showing anesthetic hysteresis, while using a PK/PD model with a slow ke0 resulted in similar CePs at LoR and RoR, thus showing no anesthetic hysteresis [27]. Thus, a difference in kinetics of drug equilibration [30] and subsequent different timings for CeP to reach the target concentration may explain why hysteresis was significantly observed in the Schnider model group [27] but obscured in the group in which it was adopted as an effect site equilibration model that readily collapses hysteresis [30].

Different aspects should be considered when explaining a difference in the occurrence of unwanted events during anesthesia between the Schnider and Eleveld model groups. A given V1 [2,3] associated with a given ke0 [27,28] may affect the initial propofol dose delivered, as observed in a computer simulation [31], during anesthesia induction with incremental doses [3]. When the CeP is increased, the TCI system briefly increases the plasma concentration to an optimal level above the target effect site concentration before temporarily stopping the infusion to allow the plasma concentration to decrease to the level of the target CeP [3]. The magnitude of the optimal plasma concentration overshoot—the peak plasma concentration that generates a gradient sufficient to cause the most rapid increase in effect site concentration but without overshooting the effect site concentration above its target—depends on the PK model [3] and critically on ke0 [3,28]. In a PK/PD model with a high V1 associated with a low ke0, compared with a PK/PD model with a low V1 associated with a high ke0 [2,3], a larger initial plasma concentration overshoot is needed to produce a greater concentration gradient between the plasma and effect site and, thus, hasten their equilibrium [3,28]. This may result in a higher risk of propofol overdose during induction of anesthesia with potentially adverse clinical consequences [28].

The Eleveld TCI model also showed a higher bias in Predictive Median Performance Error (MdPE) than the Schnider one in older patients, probably contributing to higher DAEs and BSEs in older subjects [5]. Despite aging producing alterations in cardiovascular physiology that increase the onset time of anesthetic drugs [32], an increase in sensitivity to the hypnotic drug in the elderly compared to adults [11] may exacerbate the effect of equilibration between plasma and effect site concentration and favor deep anesthesia events during TIVA-TCI in elderly patients, in particular BSEs [33], which deserve attention due to being related to postoperative delirium and neurocognitive disorders [21,34]—although this is still debated [35,36,37]. Furthermore, aging was shown to impact the depth of anesthesia monitoring and EEG-based monitoring systems; different studies have shown higher indices in elders than in young people under a comparable anesthetic plan [33,38]. Modulating the rate of propofol infusion after anesthesia induction on the basis of a difference between the expected and measured BIS, as in the Eleveld model which uses BIS as the measure of effect [2], may increase the risk of hypnotic overdosage and result in deep anesthesia events, particularly in elderly patients [33].

Eleveld TCI models should be used by anesthesiologists who are confident with this PK/PD model and knowledgeable about the PK/PD differences compared to the Schnider TCI model. When using the Eleveld TCI model for older patients, induction and initial anesthesia maintenance should be carried out in a stepwise manner with smaller variations in CePs compared to those in the Schnider TCI model. These adjustments should be guided by depth anesthesia monitoring.

This study has some limitations. First, only female patients undergoing breast surgery were involved in this observational study. Future studies should confirm our findings in a larger population of patients, including male patients and those who have undergone other types of surgery, and in an evaluation of the whole raw EEG and not only processed EEG information, particularly for the BSE evaluation.

Second, the role of remifentanil and SPI should be considered. Unconsciousness was maintained primarily by using a single hypnotic agent, such as propofol, that was titrated to recommended BIS values. However, even if remifentanil is not hypnotic, it may contribute to unconsciousness by suppressing nociceptive-induced responsiveness or arousal [39]. Monitoring the levels of both hypnosis and antinociception was shown to reduce episodes of inadequate anesthesia [7]; however, in relatively young and healthy patients undergoing elective surgery under TIVA-TCI guided by a processed EEG index and SPI, in comparison to standard monitoring alone, the reduction in unwanted anesthesia events could not be validated [8].

## 5. Conclusions

The Schnider and Eleveld models impact CePs differently, and the incidence of unwanted events during anesthesia varies. Since the Eleveld model showed a greater bias than the Schnider model in older subjects [5] and, in our study, the incidence of BSEs in elderly patients was higher in the Eleveld than in the Schnider model group, particular attention should be reserved for such a population of patients undergoing TIVA-TCI using the Eleveld model, and any phase of anesthesia should be guided by EEG monitoring, including raw waveforms and spectrograms.

## Figures and Tables

**Figure 1 life-13-02065-f001:**
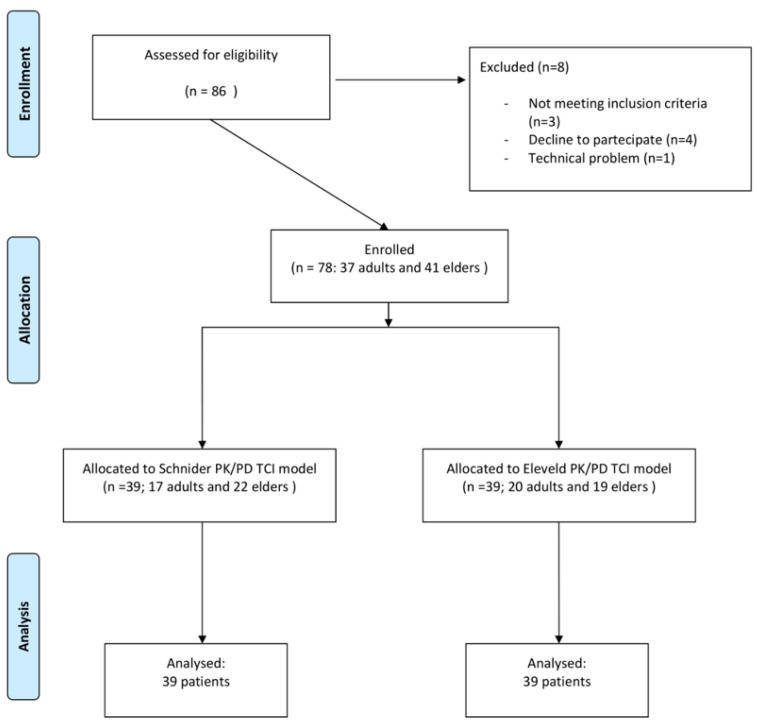
STROBE flow diagram.

**Figure 2 life-13-02065-f002:**
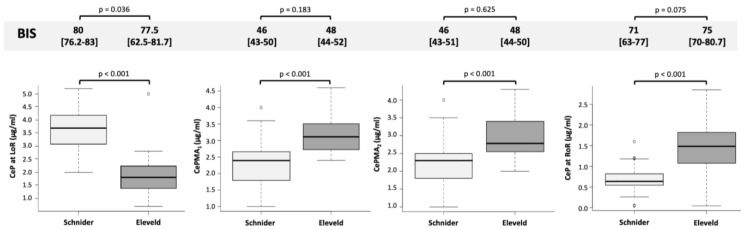
Comparison of CePs between the Eleveld and the Schnider PK/PD models across four time points during BIS-guided TIVA-TCI anesthesia in the total population of patients.

**Table 1 life-13-02065-t001:** Comparison between Schnider PK-PD model group and Eleveld PK-PD model group in total population of patients, adults, and elders.

Variable	Total Population			Adults (18–64 yrs)			Elders (≥65 yrs)		
	Schnider 39 Patients	Eleveld 39 Patients	*p*-Value	Schnider 17 Patients	Eleveld 20 Patients	*p*-Value	Schnider 22 Patients	Eleveld 19 Patients	*p*-Value
Age, yrs	66 [61–71]	63 [58–71]	0.631	60 [52–62]	58 [54–60]	0.625	69 [67–73.7]	71 [69–75.5]	0.487
Age ≥ 65 yrs, n (%)	22 (56.4)	19 (48.7)	0.650	0 (0)	0 (0)	1.00	22 (100)	19 (100)	1.00
Weight, kg	66 [61–75]	65 [57–70]	0.509	66 [58–71]	62 [56.7–67.2]	0.352	65 [62–77]	69 [59–74]	0.979
Height, cm	165 [160–168]	164 [160–167.5]	0.609	166 [160–170]	164 [160–170]	0.609	164 [160–167.7]	164 [160–165.5]	0.753
BMI, kg/m^2^	24.2 [22.6–27.6]	23.9 [21.2–27.3]	0.586	22 [22.5–27.4]	23.7 [20.2–25.1]	0.657	25.5 [23–28.8]	24.4 [23.1–28]	0.764
BMI ≥ 30, n (%)	4 (10.3)	5 (12.8)	1.00	0 (0.0)	2 (10)	0.489	4 (18.2)	3 (15.8)	1.00
ASA, n (%)									
I	9 (23.1)	7 (17.9)		9 (52.9)	7 (35)		0 (0.0)	0 (0.0)	
II	30 (74.4)	32 (74.4)		8 (47.1)	12 (60)		21 (95.5)	17 (89.5)	
III	1 (2.6)	3 (7.7)		0 (0)	1 (5)		1 (4.5)	2 (10.5)	
Propofol total dose, mg	522 [399.5–600.7]	512.6 [406.3–631.3]	0.893	527.2 [422.8–628.1]	502 [454.5–666.4]	0.831	508.8 [398.2–572.4]	533.4 [378.9–571.7]	0.937
Surgery time, min	44 [35–60]	45 [33–63.5]	0.853	43 [34–84]	48 [34.7–71.2]	0.891	46 [37.7–56.2]	44 [33–51]	0.488
Anesthesia time, min	62 [55–87.5]	66 [53–78]	0.764	60 [55–110]	80 [57.5–98]	0.795	63.5 [60.2–77.2]	65 [51–73.5]	0.425
**LoR**									
BIS baseline	97 [97–98]	97 [97–98]	0.727	97 [97–98]	98 [97–98]	0.213	97.5 [96–98]	97 [96.5–98]	0.620
CeP at LoR, μg/mL	3.60 [3.00–4.18]	1.7 [1.36–2.25]	**<0.001**	3.50 [2.65–4.07]	1.88 [1.38–2.32]	**0.001**	3.70 [3.06–4.27]	1.6 [1.29–1.99]	**<0.001**
CeR at LoR, ng/mL	0.8 [0.8–0.8]	0.8 [0.8–0.8]	1.00	0.8 [0.8–0.8]	0.8 [0.8–0.8]	1.00	0.8 [0.8–0.8]	0.8 [0.8–0.8]	1.00
BIS at LoR	80 [76.2–83]	77.5 [62.5–81.7]	**0.036**	80 [77–84]	73 [62–79.5]	**0.036**	80 [76–83]	79.5 [72.2–82.7]	0.641
**Anesthesia maintenance**									
CePMA1, μg/mL	2.40 [1.80–2.65]	3.10 [2.75–3.50]	**<0.001**	2.20 [2.20–2.70]	2.95 [2.88–3.28]	**<0.001**	2.40 [1.80–2.50]	3.30 [2.65–3.50]	**<0.001**
CeRMA1, ng/mL	3.00 [3.00–3.00]	3.00 [3.00–3.00]	0.519	3.00 [3.00–3.00]	3.00 [3.00–3.00]	0.326	3.00 [3.00–3.00]	3.00 [3.00–3.00]	0.879
BIS at CePMA1	46 [43–50]	48 [44–52]	0.183	48 [43–50]	48 [45.7–50.5]	0.324	46 [43.2–49.5]	48 [44–52]	0.598
Time to CePMA1, min	24 [20–31.5]	22 [18.5–28.5]	0.246	25 [21–33]	20.5 [19–29.2]	0.418	23.5 [20–30.5]	22 [17–27.5]	0.440
CePMA2, μg/mL	2.30 [1.80–2.50]	2.80 [2.55–3.40]	**<0.001**	2.20 [2.20–2.60]	2.90 [2.68–3.28]	**0.001**	2.40 [1.80–2.50]	2.70 [2.40–3.40]	**0.003**
CeRMA2, ng/mL	3.00 [3.00–3.00]	3.00 [3.00–3.00]	0.535	3.00 [3.00–3.00]	3.00 [3.00–3.00]	0.484	3.00 [3.00–3.00]	3.00 [3.00–3.00]	0.916
BIS at CePMA2	46 [43–51]	48 [44–50]	0.625	48 [43–50]	48 [45.7–48.5]	0.665	46 [44–51]	47 [44–50]	0.906
**RoR**									
CeP at RoR, μg/mL	0.64 [0.55–0.81]	1.48 [1.08–1.80]	**<0.001**	0.75 [0.61–0.84]	1.48 [1.21–1.76]	**<0.001**	0.60 [0.53–0.73]	1.44 [1.08–1.95]	**<0.001**
CeR at RoR, ng/mL	0.59 [0.50–0.70]	0.60 [0.50–0.77]	0.637	0.50 [0.50–0.67]	0.59 [0.49–0.76]	0.794	0.59 [0.50–0.72]	0.65 [0.50–0.88]	0.669
BIS at RoR	71 [63–77]	75 [70–80.7]	0.075	66 [60–74]	75 [69.7–82]	**0.049**	72 [70–77.7]	75.5 [70.2–78]	0.531
Time to RoR, min	9 [7–12]	10 [8–12]	0.116	8 [7–10]	11 [9–13.5]	**0.047**	9 [7–12]	9 [8–11]	0.958
Δ CeP, μg/mL	2.84 [2.28–3.48]	0.58 [0.26–0.98]	**<0.001**	2.65 [2.02–2.87]	0.58 [0.21–1.02]	**<0.001**	3.04 [2.56–3.56]	0.58 [0.33–0.98]	**<0.001**
**Unwanted events**									
DAE, n (%)	12 (30.8)	27 (69.2)	**0.001**	6 (35.3)	15 (75)	**0.022**	6 (27.3)	12 (63.2)	**0.030**
BSE, n (%)	2 (5.1)	11 (28.2)	**0.013**	1 (5.9)	5 (25)	0.189	1 (4.5)	6 (31.6)	**0.036**
LAE, n (%)	1 (2.6)	2 (5.1)	1.00	0 (0.0)	2 (10)	0.489	1 (4.5)	0 (0.0)	1.00
USRE, n (%)	1 (2.6)	2 (5.1)	1.00	0 (0.0)	2 (10)	0.489	1 (4.5)	0 (0.0)	1.00

BMI: body mass index; ASA: American Society of Anesthesiologists physical status classification; BIS: bispectral index; CeP: concentrations at the effect site (Ce) of propofol; LoR: loss of responsiveness; CePMA1: initial CeP during maintenance of anesthesia (MA); CePMA2: final CeP during MA; RoR: return of responsiveness; Δ CeP: difference between CeP LoR and CeP ROR; LAE: lightening of anesthesia event; USRE: unwanted spontaneous responsiveness event; DAE: deepening of anesthesia event; BSE: burst suppression event. The bold identifies the different phases of anesthesia or the adverse events section, or the significant variables.

## Data Availability

Data is unavailable due to privacy restrictions.

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
