# Peer review of "Schnider and Eleveld Models for Propofol Target-Controlled Infusion Anesthesia: A Clinical Comparison"

_life, 2023, doi:10.3390/life13102065_

Round 1

Reviewer 1 Report

Overall, this study is well-designed and the manuscript is well-written. Here are some suggestions:

Clarify the Inclusion of Female Patients Undergoing Oncologic Breast Surgery: In the introduction or methods section, briefly mention the rationale for including female patients undergoing oncologic breast surgery. Explain if there was a specific clinical or research-related reason for their inclusion in the study.

Provide Details About the Study Institute: In the methods section or the introduction, describe the study institute or center where the research took place. Include relevant information about its location, size, specialization, or any unique aspects that might be relevant to the study.

Elaborate on the Patient Selection Process: In the methods section, provide a more detailed description of the patient selection process. You can explain the inclusion and exclusion criteria comprehensively, including why certain exclusion criteria were chosen. For instance, if there were many exclusion criteria but only a few patients were excluded for not meeting the inclusion criteria, clarify why these specific inclusion criteria were essential to the study.

Discuss Clinical Implications: In the discussion section, add a paragraph or subsection that discusses the clinical implications of the study's findings. Explain how the results could impact clinical practice or patient care. If there are any recommendations or insights based on the findings, include them here.

Author Response

Overall, this study is well-designed and the manuscript is well-written.

Thank you, we are pleased with the favorable opinion. We have appreciated this comment.

Here are some suggestions:

Q1. Clarify the Inclusion of Female Patients Undergoing Oncologic Breast Surgery: In the introduction or methods section, briefly mention the rationale for including female patients undergoing oncologic breast surgery. Explain if there was a specific clinical or research-related reason for their inclusion in the study.

R1. Thank you. We selected this population for both surgical and anesthesiological considerations. Oncologic breast surgery, which often includes quadrantectomy or total mastectomy, is a well-standardized procedure typically performed on younger female patients with fewer health issues and a lower risk of bleeding, hypotension, and hemodynamic instability compared to other surgeries in our hospital. These aspects helped to minimize selection bias. Furthermore, anesthesia for breast surgery can be administered using supraglottic devices for airway management and mechanical ventilation under general anesthesia without the necessity of neuromuscular blocking agents. Studies have indicated that these agents can interfere with BIS values (PMID: 12873942, PMID: 26174308), potentially leading to an underestimation of adverse event detection, especially unwanted spontaneous responsiveness events - USRE, in our study. We have elaborated on this concept in the methods section.

Q2. Provide Details About the Study Institute: In the methods section or the introduction, describe the study institute or center where the research took place. Include relevant information about its location, size, specialization, or any unique aspects that might be relevant to the study.

R2. Thank you. We provided the details required.

Q3. Elaborate on the Patient Selection Process: In the methods section, provide a more detailed description of the patient selection process. You can explain the inclusion and exclusion criteria comprehensively, including why certain exclusion criteria were chosen. For instance, if there were many exclusion criteria but only a few patients were excluded for not meeting the inclusion criteria, clarify why these specific inclusion criteria were essential to the study.

R3. Thank you. We have modified accordingly.

Q4. Discuss Clinical Implications: In the discussion section, add a paragraph or subsection that discusses the clinical implications of the study's findings. Explain how the results could impact clinical practice or patient care. If there are any recommendations or insights based on the findings, include them here.

R4. Thank you. The main text discusses the clinical implications of our findings.

Reviewer 2 Report

Dear Authors,

I was interested in reviewing your manuscript. It surprised me since the title, literature presents fewer and fewer studies on pharmacokinetics of anesthetic agents in last years and your effort should be rewarded. This topic finds my complete approval for novelty and design.

Introduction

Your presentation is synthetic enough and has a good quality. Main differences between two pharmacokinetic models are adequate underlined and intelligible.

Methods

Study design is appropriate. All necessary standards were met. Your approach is statistically correct since sample size calculation.

I suggest you define your subgroups (adults vs elderly) in the text and in the STROBE flow chart. Without this definition, reading results there's a little bit of confusion evaluating populations.

Some abbreviations must be reviewed because they are not immediate during reading, and frequently the reader needs to review their meaning to catch your message. This difficulty is present in all the paper, but along the text it is weaker. I can suggest a summary table containing all abbreviations.

All statistical tests are applied correctly considering the sample size and characteristics of variables. You also used a refined tool like AIC.   

Results

Exposition of results is clear and concise. Tables are rich of numbers but intelligible and correctly commented in the text. 

I have no significant suggestion.

Discussion

As for the rest of the manuscript, I found your discussion adequate but to concise. Even if your study considers a small sample size (but adequately supported by statistical analysis), results of your comparison are important for clinical uses, and I was hoping to find some clinical suggestion for propofol infusion with both pharmacokinetic models and also a graphical abstract could upgrade visibility of this work, focusing on your title "a clinical comparison".

I thank you for this occasion.

I hope that my suggestion could improve your work.

Kind Regards

Lorenzo

I'm not a native English, so my evaluation could not be complete, but sometimes sentences are too long and complex, and they require a careful reading.

I suggest minor english revision.

Author Response

I was interested in reviewing your manuscript. It surprised me since the title, literature presents fewer and fewer studies on pharmacokinetics of anesthetic agents in last years and your effort should be rewarded. This topic finds my complete approval for novelty and design.

Thank you. We're delighted to receive this positive feedback.

Introduction

Your presentation is synthetic enough and has a good quality. Main differences between two pharmacokinetic models are adequate underlined and intelligible.

Thank you. We greatly value this positive comment.

Methods

Study design is appropriate. All necessary standards were met. Your approach is statistically correct since sample size calculation.

Thank you. We are pleased to have met the criteria for a valuable study.

Q1. I suggest you define your subgroups (adults vs elderly) in the text and in the STROBE flow chart. Without this definition, reading results there's a little bit of confusion evaluating populations.

R1. Thank you for your suggestion. We made the necessary changes.

Q2. Some abbreviations must be reviewed because they are not immediate during reading, and frequently the reader needs to review their meaning to catch your message. This difficulty is present in all the paper, but along the text it is weaker. I can suggest a summary table containing all abbreviations.

R2. Thank you. We modified accordingly.

All statistical tests are applied correctly considering the sample size and characteristics of variables. You also used a refined tool like AIC.

Thank you. We are pleased about it.

Results

Exposition of results is clear and concise. Tables are rich of numbers but intelligible and correctly commented in the text. I have no significant suggestion.

Thank you. We are pleased to have met the reviewer's expectations.

Discussion

R3. As for the rest of the manuscript, I found your discussion adequate but to concise. Even if your study considers a small sample size (but adequately supported by statistical analysis), results of your comparison are important for clinical uses, and I was hoping to find some clinical suggestion for propofol infusion with both pharmacokinetic models and also a graphical abstract could upgrade visibility of this work, focusing on your title "a clinical comparison".

Q3. Thank you. The main text covers the clinical implications of our findings. We've included a graphical abstract to summarize our key results and offer clinical suggestions for propofol infusion using both pharmacokinetic models.
